# Using Generic Direct M-SVM Model Improved by Kohonen Map and Dempster–Shafer Theory to Enhance Power Transformers Diagnostic

**Mounia Hendel [1], Fethi Meghnefi [2], Mohamed El Amine Senoussaoui [3] , Issouf Fofana [2,*] and Mostefa Brahami [4]**

[1] LGEM, Ecole Supérieure en Génie Electrique et Energétique d'Oran, Oran 31000, Algeria
[2] Canada Research Chair, Tier 1, ViAHT, University Québec, Chicoutimi, QC G7H 2B1, Canada
[3] LGPCS, University Mustapha Stambouli of Mascara, Mascara 29000, Algeria
[4] ICEPS, Faculty of Electrical Engineering, Djilali Liabes University of Sidi Bel Abbes,
    Sidi Bel Abbes 22000, Algeria; mbrahami@yahoo.com
[*] Correspondence: ifofana@uqac.ca

**Abstract:** Many power transformers throughout the world are nearing or have gone beyond their theoretical design life. Since these important assets represent approximately 60% of the cost of the substation, monitoring their condition is necessary. Condition monitoring helps in the decision to perform timely maintenance, to replace equipment or extend its life after evaluating if it is degraded. The challenge is to prolong its residual life as much as possible. Dissolved Gas Analysis (DGA) is a well-established strategy to warn of fault onset and to monitor the transformer's status. This paper proposes a new intelligent system based on DGA; the aim being, on the one hand, to overcome the conventional method weaknesses; and, on the other hand, to improve the transformer diagnosis efficiency by using a four-step powerful artificial intelligence method. (1) Six descriptor sets were built and then improved by the proposed feature reduction approach. Indeed, these six sets are combined and presented to a Kohonen map (KSOM), to cluster the similar descriptors. An averaging process was then applied to the grouped data, to reduce feature dimensionality and to preserve the complete information. (2) For the first time, four direct Multiclass Support Vector Machines (M-SVM) were introduced on the Generic Model basis; each one received the KSOM outputs. (3) Dempster–Shafer fusion was applied to the nine membership probabilities returned by the four M-SVM, to improve the accuracy and to support decision making. (4) An output post-processing approach was suggested to overcome the contradictory evidence problem. The achieved AUROC and sensitivity average percentages of 98.78–95.19% (*p*-value < 0.001), respectively, highlight the remarkable proposed system performance, bringing a new insight to DGA analysis.

**Keywords:** DGA; probabilistic M-SVM; Generic M-SVM Model; Dempster–Shafer Rule; Kohonen Map

## 1. Introduction

Indicators of sustainability focusing on energy are crucial tools used to assess and monitor progress toward guaranteeing electricity delivery to end-users [1]. In the last decades, power grids have been facing growing interest in deploying new and intelligent technologies to obtain improved reliability and availability of power supply. This is important in meeting new challenges due to accelerating urbanization and evolving requirements to ensure smart cities. The smart city concept mostly relies on cameras, sensors and monitoring tools to maintain or support human well-being continuously over time. The data collected are processed and analyzed to improve operational efficiency of major equipment such as power transformers, public safety, life quality, and also ensure efficient

electrical installations [2,3]. In this context, accurate monitoring of major assets is essential. Power transformers, which are the most essential and expensive devices in the power transmission and distribution networks, are aging worldwide ahead of their theoretical design life. Their role is to facilitate the transition between the different electrical network levels (production, transport and distribution), while minimizing losses from the Joule effect. Due to their importance in an electrical structure, their reliable operation guarantees a good efficiency to the entire distribution network. In particular, when a power transformer suddenly explodes due to various factors, it may affect the generator output, which causes significant damage to the electric company's economy and to the user's property, and may also cause irreversible damage to human safety, especially to maintenance personnel. Thus, it is necessary to ensure an excellent monitoring of the power transformer's condition in order to avoid sudden catastrophic failures [4,5].

Throughout their operation, power transformers are continuously subjected to different stresses (electrical, thermal and mechanical) and aggressive chemical byproducts. Excessive transformer stresses adversely impact insulating materials (solid insulation and oil), leading to an internal transformer failure [6]. This failure can be observed by analyzing the combustible and non-combustible gases formed and dissolved in the transformer's oil, such as: Hydrogen ($H_2$), Oxygen ($O_2$), Nitrogen ($N_2$), Carbon Monoxide (CO), Methane ($CH_4$), Carbon Dioxide ($CO_2$), Ethylene ($C_2H_4$), Ethane ($C_2H_6$), Acetylene ($C_2H_2$), etc. It is therefore essential to regularly analyze the oil's condition to detect potential defects. This is done by measuring the dissolved gas concentrations [1,7]. To date, several strategies have been proposed for transformers condition monitoring from oil (Furan analysis, Physico-chemical analyses, Dissolved gas analysis (DGA), etc.). Among these, DGA is the most widely used approach for power transformer diagnosis, since it is non-invasive, very simple to implement, inexpensive and above all effective.

In the last decades, a large number of DGA-based approaches have been proposed. These approaches can be subdivided into two categories: traditional methods and Artificial Intelligence (AI)-based techniques. Traditional methods mainly include: Duval's Triangle (D-T) and Pentagons [8], Rogers Ratios (R-R) [9], Dornenburg Ratios (D-R) [10], IEC Ratios (IEC-R) [10] and Key Gases (K-G) [11]. However, conventional DGA methods mainly perform diagnosis by coding, which leads to problems such as absolute code limits, missing codes and sensitivity to gas volume fraction fluctuations, etc. For example, the IEC-R precision is of concern for incomplete coding reasons (it is, for example, impossible to differentiate between low- and high-energy discharge). R-R are only effective for detecting thermal faults and are ineffective for detecting other faults. The D-R generally detects three fault categories and is not able to return the fault severity. D-T is mainly based on the use of five triangles, and triangles 4 and 5 are in several cases contradictory [7]. The reason why the different approaches can give contradictory answers for the same sample is unclear, and it is difficult for the expert to prioritize an answer. This can induce misinterpretation, leading to incomplete and/or uncertain classification. Hence, there is a need to find powerful alternatives that are not limited by the programmer's intelligence and that are able to learn and to improve by examples and face situations never seen before, such as AI alternatives. Indeed, these methods are mainly based on the definition of one/or more decision boundaries based on very powerful mathematical models and take into account a sample global (non-partial) exploration. In this sense, several ingenious studies based on AI approaches have been proposed (separately or combined with conventional methods) [1,7,12–14] and have proven their effectiveness in comparison to conventional methods.

In this contribution, a new approach for monitoring the power transformer's condition is presented. The approach is based on a two-stage hybrid system: a descriptor extraction and construction stage, and a discrimination and outputs merging stage. The block diagram of the proposed Power Transformer Diagnosis (PTD) approach is depicted in Figure 1.

**Features extraction and reconstruction stage**

**Figure 1.** Block diagram of the proposed PTD approach.

- In the first stage (Stage-1), a relevant descriptor space is constructed. Six descriptor sets extracted according to six distinct DGA techniques are firstly retained: K-G, IEC-R, R-R, D-R, D-T and Gases Percentage (G-P). These parameters are then combined. This is followed by a redundant parameters elimination phase. After that, a standardization process is considered to formalize all the parameters to the same interval. Finally, a novel descriptor reconstruction stage based on the Kohonen Self-Organizing Maps (KSOM) [15] is proposed to decrease the data model and facilitate the discriminator's work while keeping all the information.
- In the second stage (Stage-2), four direct probabilistic Multiclass Support Vector Machines (M-SVM) are implemented for the first time, via a Generic M-SVM Model (GM-SVM) [16]: the Weston and Watkins (WW) model [17], the Crammer and Singer (CS) model [18], the Lee et al. (LLW) model [19], and the Quadratic Loss Multi-Class Support Vector machine (M-SVM$^2$) model [20]. These four models are used as a solution to overcome the weaknesses of the indirect M-SVM models widely used in this problem. Each used M-SVM considers the returned parameters set by Stage-1, and calculates the belonging probabilities to each of the considered nine classes.
- Then, to approve the final outputs, the Dempster–Shafer (DS) [21,22] fusion is applied to combine the four M-SVM outputs within the beliefs and evidence model framework. Also, this stage proposes an alternative to post-process the after-fusion outputs: the example is considered well-classified if its probability exceeds a decision threshold; otherwise, the example is assigned to the reject class.

Thus, the proposed system has a primary objective to improve the descriptor extraction step with the proposed KSOM parameters reconstruction approach. The latter effectively minimizes the information loss as much as possible, unlike the traditional function-

alities selection approaches. The second objective focuses on implementing the four direct M-SVM through the Generic M-SVM Model, with the aim of overcoming the limits of traditional M-SVM based on decomposition methods (minimize the classifier complexity and save execution time). Finally, the proposed system has the final objective of strengthening the final decision making by merging four M-SVM by DS Fusion. After that, an approach to solve the contradictory evidence problem, linked to DS fusion, is proposed. Section 2 describes in detail the motivations and the originality of the approach taken.

The paper is organized as follows. Section 2 exposes the foundations, the motivations and the innovations of the study. Section 3 explains the adopted parameter extraction approaches, as well as the considered data reconstruction approach. Section 4 exposes the theoretical framework of probabilistic direct M-SVM and the theoretical framework of the DS fusion model. Section 5 presents the used database description, the M-SVM hyper parameters selection, plus the performance enumeration and their statistical analysis. The paper is terminated with a conclusion and perspectives in Section 6.

## 2. Study Motivations and Innovations

One of the key points of a PTD method lies in the dissolved gases representation domain assortment (choice) through the parameter extraction. This is due to the fact that the discriminator generalization strongly depends on the parameter space. Different representations have been exposed in the literature, such as: K-G [7,14,21–24], IEC-R [14,25,26], Personalized Ratios [14,25], R-R [25,26], Logarithmic Data Transformation [23], G-P [23,26], DGA code [27], Standardized Data [23], D-R [26], D-T [25], etc. Also, several comparative studies between these descriptors [23–26] have been carried out, but each of these approaches has advantages and disadvantages [7]; it would be more interesting to take the benefits of each one through a combination.

However, PTD applications are often characterized by a low learning vector number, and a large vector dimension necessarily accentuates the curse of dimensionality. Indeed, the phenomenon is induced by the fact that there is a strong dissimilarity between the observations and a great divergence among the training examples. This lack of data density inevitably affects and alters the discriminator generalization, whose foundation is essentially based on statistical significance. The dimensionality reduction appears as the main reflection in the face of the curse of dimensionality. There are two categories of dimensionality reduction approach: the variable selection approaches, which consist of electing a descriptor sample from the global variables set; and the variable transformation approaches, which consist of reconstructing a new descriptor set based on the similarity characteristics of the initial variables. The first method category generates a loss of useful information (partial data exploitation), which makes it suboptimal compared to the second [28,29].

For dimensionality reduction, few previous works have been proposed, and to the best of our knowledge, the work carried out in this context is based on the descriptor selection approaches. For example, the authors in [14] proposed a Practical Swarm Optimization (PSO) algorithm for feature selection. Xie et al. [12] proposed an approach based on relief algorithms for parameter reduction. Finally, one study [30] proposed a parameter selection method based on the Genetic Algorithm (GA). The current paper first exposes the use, for the first time, of a very powerful AI approach: KSOM, for merging and reconstructing six descriptor sets that are the most used in real applications (K-G, IEC-R, R-R, D-R, G-P, D-T); the objective being to reduce the descriptor space and to preserve the data entirety contribution.

The issues related to a classification method choice are also very important. The choice must be related to several parameters, more particularly: the nature of the data (noisy, overlapping, redundant, and/or disordered) and the learning set size and nature (small or large). Generally, in PTD applications one encounters non-linear data, a very small training set (as earlier reported), non-equiprobable categories and a significant correlation between the distinct category examples. In this case, the SVM are more efficient than neu-

ral networks and offer remarkable generalization results due to their advantages. A few possibilities can be explored:

- Embedded in real-time processes (low execution time);
- Applied to randomly distributed and unknown data;
- Ensure a global optimum due to the convex optimization principle;
- Do not suffer from overfitting and over-learning problems;
- Decrease the curse of dimensionality.

Therefore, several previous studies have proposed SVM-based PTD systems [1,21,30–32].

The PTD is a multi-class discrimination problem, but SVM were induced by Cortes and Vapnik [33] to cover two-category discrimination problems. Unlike other machine learning approaches, where switching to the polytomy case is intuitive, SVM in most cases use decomposition methods (one-against-one, one-against-all, etc.). These dichotomous approaches generate significant complexity, which leads to a considerable execution time (final decision making, training algorithm), especially when dealing with large category classification problems. To overcome this problem, researchers are currently working on proposing new direct intuitive models that consider a single M-SVM for unique multi-class learning problem resolution. And to this date, four powerful models have been proposed, namely: the WW model, GS model, LLW model and M-SVM$^2$ model. Geurmeur then proposed an innovative GM-SVM model, which brings together the four direct M-SVM offered to date. This research presents, as a second contribution, the introduction of the four direct M-SVM models in the PTD application via the GM-SVM model.

Also, serval information sources fusion is strongly recommended in discrimination problems, especially if false prediction stakes exert influence in a consistent way, such as in the case of this application. It is indeed much more informative to make a decision knowing that several sources agree. This study fits into the evidence theory context, which takes into consideration ambiguity and vagueness, credibility and the conflict of different independent proofs, and then produces a certain and a confident result. A number of previous works have studied DS fusion, whether to interpret the relationship between different data sources (DGA [14], Ratio methods [28], etc.), or the merging of several discriminator outputs [1,7,31,32], and have highlighted the theory contribution in the PTD problem.

For DS theory, when the classifiers are in perfect agreement or the conflict between them is weak, confidence in the final decision is reinforced by a higher recognition rate. However, if the discriminators are in strong conflict, the final decision is translated into a random result, which is necessarily questioned (one of the major DS law criticisms). To overcome this problem, the authors in [32] proposed to symbolize the reliability differences between the discriminator outputs by weighting coefficients; the latter allow reconstructing the base probabilities according to the attributed priority to the proofs. Nevertheless, the four direct M-SVM models implemented in this contribution offer theoretically the same performances, and each model has its own advantages and disadvantages. It is then practically impossible to prioritize one model. On the contrary, it would be more interesting to exploit the performances of each of them in order to reconstruct a more robust model—hence the interest in the merger. The third contribution of this study consists of the DS fusion of four direct M-SVM and proposition of an output post-processing approach to solve the contradictory evidence problem.

Thus, by implementing the proposed intelligent fault diagnosis system, it is hoped to:

- Increase the four M-SVM performance by facilitating their classification task, thanks to the proposed descriptors reconstruction approach;
- Reduce complexity and save execution time by implementing direct M-SVM instead of M-SVM based on decomposition methods;
- Strengthen M-SVM outputs by translating them into posterior probabilities;
- Strengthen decision-making, gain sensitivity and minimize false alarms by applying the DS fusion and the rejection class introduction.

## 3. Features Extraction and Reconstruction Methods

This section defines the adopted methodology for the input model construction (of the four direct M-SVM from DGA samples, which includes the following steps:

### 3.1. Feature Extraction Approaches

The literature exposes a multitude of interpretative-based DGA approaches to determine the condition of oil-immersed transformers. In this contribution, the following six approaches are considered:

### 3.1.1. Key Gases (K-G)

The first data set is built from nine type of gases: $H_2$, $O_2$, $N_2$, $CO$, $CH_4$, $CO_2$, $C_2H_4$, $C_2H_6$, $C_2H_2$. The latter intervene in an effective way for the considered power transformer fault interpretation, according to IEC 60599 standard [23].

### 3.1.2. Gases Percentage (G-P)

The second set is based on nine gas percentage concentrations (retained in the first set): $\%H_2$, $\%O_2$, $\%N_2$, $\%CO$, $\%CH_4$, $\%CO_2$, $\%C_2H_4$, $\%C_2H_6$, $\%C_2H_2$; relative to the total combustible and non-combustible gases sum (TG): $\sum(H_2, O_2, N_2, CO, CH_4, CO_2, C_2H_4, C_2H_6, C_2H_2)$.

### 3.1.3. IEC Ratios (IEC-R)

The third set of data counts three ratios, which are generated from five key gases as follows:

$$R1 = {}^{C_2H_2}/_{C_2H_4} \tag{1}$$

$$R2 = {}^{CH_4}/_{H_2} \tag{2}$$

$$R3 = {}^{C_2H_4}/_{C_2H_6} \tag{3}$$

### 3.1.4. Rogers Ratios (R-R)

The fourth descriptors set that is used considers the same ratios as those of the IEC method, in addition to the following ratio:

$$R4 = {}^{C_2H_6}/_{CH_4} \tag{4}$$

### 3.1.5. Dornenburg Ratios (D-R)

The fifth features set is a vector with four elements $R_1$ and $R_2$ of the IEC method, plus the two following ratios:

$$R3 = {}^{C_2H_2}/_{CH_4} \tag{5}$$

$$R4 = {}^{C_2H_6}/_{C_2H_2} \tag{6}$$

### 3.1.6. Duval's Triangle (D-T)

The last set of descriptors is expressed as a percentage of three key $\%CH_4$, $\%C_2H_2$, and $\%C_2H_4$, relative to their total concentration (TS): $(CH_4, C_2H_2, C_2H_4)$. The reader is invited to see reference [28] for more details on the graphing approach via Duval's Triangle 1.

### 3.2. Features Combination and Redundant Features Elimination

Once all six features are defined by the selected diagnostic criteria application, a concatenation process is applied to generated vectors. This allows obtaining a global vector made up of 32 descriptors. The redundant parameters are then eliminated in order to produce a final vector composed of 27 parameters; these are considered input features to a KSOM map.

### 3.3. Features Standardization

For the purpose of an efficient preparation of discriminator inputs, it is generally recommended to implement a standardization process (to ensure optimal data modeling). This consists in using a common scale for all descriptors while preserving an identical general distribution and a similar ratio to those of the original parameters. In this sense, a centered/reduced type standardization process is applied:

$$\widetilde{x_{ij}} = \frac{\left(x_{ij} - \overline{x_j}\right)}{\sigma_{xj}} \tag{7}$$

Note that $\overline{x_j}$ and $\sigma_{xj}$ represent, respectively, the mean and the standard deviation of the $j$th descriptor vector parameter.

Note: the standardization phase was carried out before the new dataset reconstruction, to facilitate the Vector Quantification (VQ) process by minimizing data dispersion.

### 3.4. Kohonen Self-Organizing Map Reduction

This research presents the implementation of an unsupervised learning SOM map (see Definition 1) to project the initial parameters space (of 27 descriptors) onto a reduced parameters space (of nine descriptors) based on the VQ principle (without information loss). The new descriptors are given by the external weights, which connect the map neurons to the input vectors. The learning consists of finding the weights which best cover the input space by applying the illustrated algorithm in Definition 1.

**Definition 1 (Kohonen map unsupervised learning problem) [15].** Considering a $M$ dimension learning set, where each learning vector consists of $N$ descriptors. Let $(x_1, x_2, \ldots, x_M)$ be a set of input vectors of N dimension. Let $v_j = (v_{j1}, v_{j2}, \ldots, x_{jN})$ be an external weights vector that connects neuron $j \in [\![1, n \times m]\!]$ to an input vector $x_i$; and let $w_{jl}$ be an internal weights vector (obtained through the "Mexican hat" function) that connects neuron j to all these neighbors $l \in [\![1, n \times m]\!] \backslash j$. The KSOM unsupervised learning occurs in three steps:

- Step 1: Elect cluster with maximum response.
  - Apply to the map input a learning vector $x_i$.
  - Determine the best matching unit $j$, whose vector $v_j$ is closest to the input $x_i$:

$$d_j(t) = \min_{1 < k < n \times m} \left( \sum_{i=1}^{N} \|x_i(t) - v_{ik}(t)\| \right)$$

  - Winning neuron j activation.
- Step 2: neighborhood construction around the winning neuron.
- Step 3: adaptation of winning cluster weights:

$$v_j(t+1) = v_j(t) + \alpha(t)\|x_i(t) - v_{ij}(t)\|$$

$$v_l(t+1) = v_l(t) + \alpha(t)\|x_i(t) - v_{il}(t)\|$$

where $(\alpha < 1)$ is the learning step which decreases according to the iterations t.

- Back to step 1 until the algorithm converges.

Figures 2 and 3 show these weights at the start and at the end of training. When the algorithm starts, the weights of all the neurons in the map are grouped and initialized to low values. During the iterations, they begin to occupy the input space; the learning ends up by taking into account all of this space. Thus, each neuron represents a part of this space.

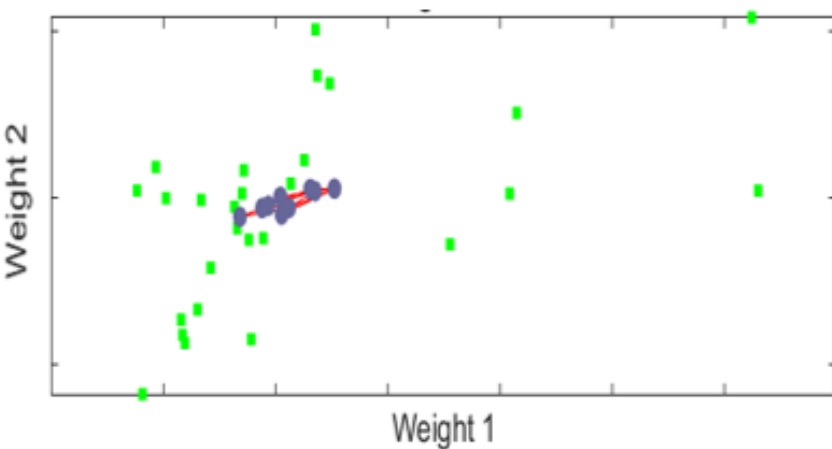

**Figure 2.** KSOM weight positions at the training start.

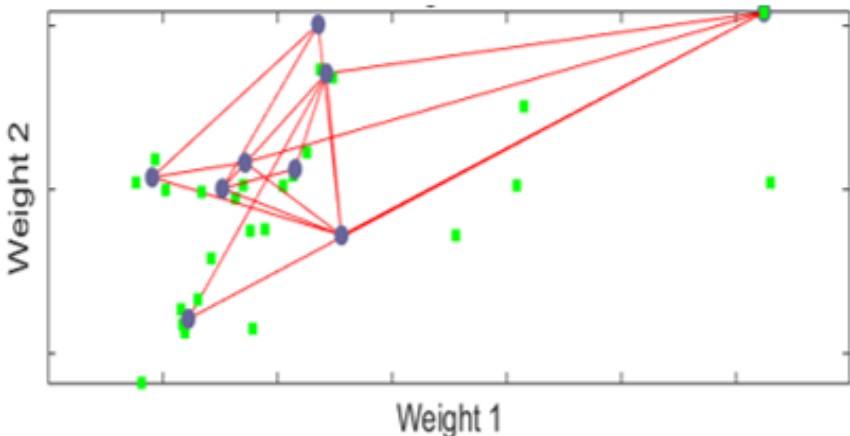

**Figure 3.** KSOM weight positions at the end of the training.

To select the retained map size, six ($n \times m$), different configurations were evaluated (with $n$ and $m \in [2, 5]$). Each of these configurations generated fifteen epochs. Each epoch was then evaluated by a MultiLayer Perceptron (MLP) to choose the most efficient among the fifteen (from a generalization rate view point). Figure 4 shows the parameter distribution around the clusters for the six selected configurations. Also, Table 1 illustrates the classification rates relating to these same topologies. It can thus be possible to report the following information:

- The minimum extreme dimension choice (the number of neurons in the map) is based on the fact that the majority of conventional methods use on average four ratios for decision-making. So a set of four parameters is to be reconstructed.
- The maximum extreme dimension choice is fixed after dead neurons' appearance, whose external connections do not represent any parameter; just as it is important to not go beyond the initial parameter number (final set < initial set).
- The final map choice ($3 \times 3$ configuration) among the six selected topologies is based on a trade-off between the neuron number and the maximum generalization rate, plus the dead neurons' absence.
- The new vector dimension is equal to the retained final map neuron number (for a total of nine parameters).
- The value of each new descriptor is obtained by averaging the parameters associated with a given cluster.

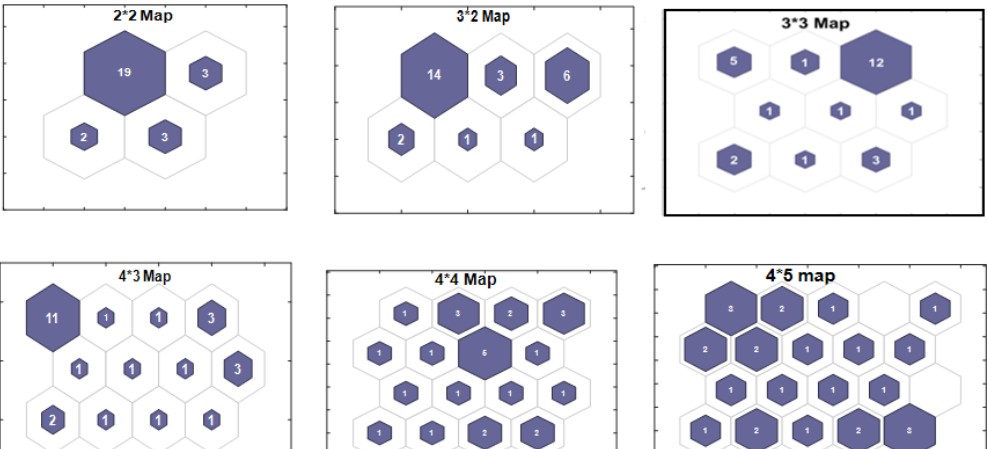

**Figure 4.** Parameter distribution around the clusters for the six selected configurations.

**Table 1.** The six retained configuration classification rates.

| Map Size | Generalization Rate (%) | Map Size | Generalization Rate (%) |
|---|---|---|---|
| 2 × 2 Map | 82.94 | 4 × 3 Map | 88.10 |
| 3 × 2 Map | 85.32 | 4 × 4 Map | 87.30 |
| 3 × 3 Map | 88.10 | 4 × 5 Map | 87.30 |

Table 2 exposes the new descriptors after learning according to the initial parameters. It can thus be possible to see that some map neurons cover only one input descriptor, for example:

- Neuron 2 (D2) represents the initial descriptor $C_2H_2/C_2H_4$. This ratio is used by the three conventional methods: IEC-R, R-R and D-R, which considers it important.
- Neuron 4 (D4) evokes the initial Dornenburg R3 report, considered very important for detecting the low-energy thermal fault category from other categories.
- Neuron 6 (D6) is seen as being a support and confirmation element of the partial discharge fault if it is greater than 30%.

**Table 2.** The new descriptors given by the card after learning.

| The Nine New Descriptors | Initial Descriptors They Represent |
|---|---|
| D1 | $H_2$, $H_2/TG$ |
| D2 | $C_2H_2/C_2H_4$ |
| D3 | $CH_4/TS$, $C_2H_2/TS$, $C_2H_4/TS$ |
| D4 | $C_2H_6/CH_4$ |
| D5 | $C_2H_2$ |
| D6 | $CH_4/H_2$ |
| D7 | $O_2$, $N_2$, $CO$, $CO_2$, $CO/TG$ |
| D8 | $CO_2/TG$ |
| D9 | $CH_4$, $C_2H_4$, $C_2H_6$, $O_2/TG$, $N_2/TG$, $CH_4/TG$, $C_2H_4/TG$, $C_2H_6/TG$, $C_2H_2/TG$, $C_2H_4/C_2H_6$, $C_2H_2/CH_4$, $C_2H_6/C_2H_2$ |

Also, other neurons cover several old descriptors, which are generally strongly correlated (information redundancy). For example:

- Neuron 1 (D1) brings together the two descriptors $H_2$, $H_2/TG$, which model the same information revealing the presence of the partial discharge type defect.

- Neuron 7 (D7) brings together five descriptors involved in an insulation fault identification.

All of these findings confirm the coherence of the groups obtained after Kohonen map application.

## 4. Discrimination and Outputs Merging Stage

In this section, the mathematical foundations of direct probabilistic M-SVM, as well as the theoretical framework of the DS merging rule, are described.

### 4.1. Direct Probabilistic Multiclass Support Vector Machines (M-SVM)

Nowadays, SVM finds applications in various fields, and those because of their advantages. The SVM do not consider multi-category classification natively; they are created to calculate dichotomies. Thus, for multi-class applications, two categories of M-SVM can be considered; the first category (indirect M-SVM) is based on the use of decomposition techniques, while the second category (direct M-SVM) is based on a multivariate affine model and currently has four models: the WW model, GS model, LLW model, and M-SVM$^2$. Geurmeur then devised a Generic GM-SVM model, which is a statistical properties unified representation of the four direct M-SVM. The GM-SVM model is considered in this paper to present in a global way the learning problem of each direct M-SVM.

#### 4.1.1. Generic M-SVM Model

Direct M-SVM belong to the class of kernel learning algorithms and are considered as a direct generalization of binary SVM for the multi-category case (without using the decomposition approaches). Therefore, direct M-SVM operate in a specific function class $H_{k,Q}$ deduced from another function $H_k$.

**Definition 2 (Class of functions $H_{k,Q}$).** Let $X = \{x_1, x_2, \ldots x_n\}$ be a set of vectors, where each vector is labelled according to $(y_i \in [\![1, Q]\!]) \in Y$, with $Q \geq 3$. Let $H$ be a Hilbert space endowed with its scalar product $\langle ., \rangle_{.H}$, and $(H_k \langle ., . \rangle_{.H_k})$ be the homologous Reproducing kernel Hilbert Space (RKHS) induced by a positive k kernel [34]. The direct M-SVM class functions associated with the kernel k are represented as follows:

$$H_{k,Q} = (H_k \oplus \{1\})^Q$$

knowing that:

$\{1\}$ : be the space of real-valued constant functions on $X$.
and

$$\forall h \in H_{k,Q}, \ \forall x \in X, \ h(x) = \overline{h}(x) + b = \left( \langle \overline{h}_k, k(x, .) \rangle_{H_k} + b_k \right)_{1 \leq k \leq Q}$$

With:

$$\overline{h} = (\overline{h}_k)_{1 \leq k \leq Q} \in H_k^Q$$

$$b = (b_k)_{1 \leq k \leq Q} \in R^Q$$

(for more details see the Definitions 1–3 provided in Section 2.1 of [34]).

**Definition 3 (Generic M-SVM model, Section 2.1 Definition 4 in [16]).** Let a learning set $z_m = \{(x_i, y_i); i = 1, m\}$ with $x_i \in X$ and $y_i \in [\![1, Q]\!]$; let $H_{k,Q}$ be the functions Class of kernel k. The multi-class direct M-SVM discriminator training consists of processing and resolving a quadratic program, which is defined as follows:

$$\min_{h, \xi} \{ \|M\xi\|_P^P + \lambda \|P_{\mathbf{H}_k} h\|_{\mathbf{H}_k}^2$$

$$s.t. \begin{cases} \forall i \in [\![1,m]\!], \ \forall k \in [\![1,Q]\!] \backslash \{y_i\}, k_1 h_{y_i}(x_i) - h_k(x_i) \geq k_2 - \xi_{(i-1)Q+k} \\ \forall i \in [\![1,m]\!], \ \forall (k,l) \in ([\![1,Q]\!] \backslash \{y_i\})^2, k_3 \left( \xi_{(i-1)Q+k} - \xi_{(i-1)Q+l} \right) = 0 \\ \forall i \in [\![1,m]\!], \ \forall k \in [\![1,Q]\!] \backslash \{y_i\}, (2-p)\xi_{(i-1)Q+k} \geq 0 \\ (1-K_1) \sum_{k=1}^{Q} h_k = 0 \end{cases}$$

where:

The bounded operator $P_{H_k}$: defines an orthogonal projection from $H_{k,Q}$ into $H_k$.

$\xi \in R^{Qm}(Z_m)$ are the slack variables, with $(\xi_{(i-1)Q+y_i})_{1 \leq i \leq m}$.

$(\lambda, \ p, K_2) \in R_+^*$ and $(K_1, K_3) \in \{0,1\}^2$.

The matrix $M[Qm, Qm]$ rank is equal to $(Q-1)_m$, and it is a diagonal matrix if

$$p = 1.$$

Let $\delta$ be the Kronecker symbol; so, let us specify the general term of $I_{Qm}$ and $(M)^2$, respectively:

$$m_{ij,jl} = \delta_{i,j}\delta_{k,l}\left(1 - \delta_{y_i,k}\right).$$

$$m_{ij,jl}^{(2)} = s\left(1 - \delta_{y_i,k}\right)\left(1 - \delta_{y_i,l}\right)\delta_{i,j}(\delta_{k,l} + \frac{\sqrt{Q}-1}{Q-1}) \tag{8}$$

Based on the Generic model unification, Table 3 illustrates the hyper parameter characteristics relating to each of the four M-SVM models.

**Table 3.** Specifications of the four M-SVM as instances of the GM-SVM model.

| M-SVM | M | p | $K_1$ | $K_2$ | $K_3$ |
|---|---|---|---|---|---|
| WW Model | $I_{Qm}Z_m$ | 1 | 1 | 1 | 0 |
| CS Model | $\frac{1}{Q-1}I_{Qm}Z_m$ | 1 | 1 | 1 | 1 |
| LLW Model | $I_{Qm}Z_m$ | 1 | 0 | $\frac{1}{Q-1}$ | 0 |
| M-SVM$^2$ | $(M)^2$ | 2 | 0 | $\frac{1}{Q-1}$ | 0 |

For this experiment implementation, an open-source software package (MSVM-pack) [35] based on the GM-SVM model is used.

### 4.1.2. Probabilistic Direct M-SVM

Power transformer automatic diagnosis is a critical and complex task; the discrimination error costs are in most cases asymmetrical, difficult to quantify and differ between experts and transformers. Thus, it is desirable and more reliable to consider classifiers that return a posteriori probability estimates, instead of a deterministic classification, which returns the membership class label (this allows associating a confidence degree with the final decision, for example knowing that the probability that the transformer presents a fault is 80% is better than not having any indicators). These probabilistic estimations can then be joined to other information sources for complex but more credible final decision making. However, SVM are at their basis deterministic discriminators. Nevertheless, it is possible to obtain a M-SVM posterior probability estimator by adapting Platt's binary case solution [36] to the multi-class case, as follows:

$$k \in [\![1,Q]\!], \widetilde{h}_k = \frac{\exp(h_k)}{\exp\left(\sum_{k=1}^{Q} h_k\right)} \tag{9}$$

### 4.2. Dempster–Shafer (DS) Theory

Developed by Dempster [21] and formalized mathematically by Shafer [22], belief theory (or evidence theory) allows imprecision and uncertainty to be modeled simultaneously based on the belief function (*Bel*) and the plausibility function (*Pl*); the latter are deduced from the mass functions (*m*). Below, the Mass, Plausibility and Belief functions, as well as the merging and the final decision rules, are explained.

#### 4.2.1. Mass, Plausibility and Belief Functions

Let $\Omega = \{\varnothing, w_1, w_2, \ldots, w_N\}$ be the set subspaces (focal elements) of the discernment frame. The liveness relative to a proposition *A* (for a given source *S*) can be quantified by a mass function "*m*" (coryance measurement) $m : 2^\Omega \to [0,1]$ and must respect the two following properties: $m(\varnothing) = 0$ and $\sum_{A \subseteq \Omega} m(A) = 1$.

From the mass function, it is easy to deduce *Bel* and *Pl* functions attributed to the focal element *A*, in accordance with the following two mathematical equations:

$$\forall A \in 2^\Omega, Bel(A) = \sum_{B \subseteq A \neq \phi} m(B) \tag{10}$$

$$\forall A \in 2^\Omega, Pl(A) = \sum_{B \cap A \neq \phi} m(B) = 1 - Bel(\overline{A}) \tag{11}$$

With $\overline{A}$ being the opposite event of the proposition *A*. The quantity $Bel(A)$ illustrates the belief strength in *A*, justified by the evidence information taken into consideration. Plausibility is the credibility dual function; its quantity $Pl(A)$ is considered as an upper bound on the belief strength likely to be attributed to *A* following new data.

#### 4.2.2. Merging Rule

The merging rule can be generally applied at three levels, namely: low level (considering the raw data), intermediate level (descriptor merging) and high level (discriminator merging). In this work, a fusion at the high level is retained.

Masses combination on the basis of DS law is achieved by Dempster's orthogonal sum; thus, for two masses functions $m_1$ and $m_2$ and for all $A \in 2\Omega$, it yields:

$$m(A) = (m_1 \oplus m_2)(A) = \frac{\sum_{B_1 \cap B_2 = A} m_1(B_1) \cdot m_2(B_2)}{1 - K} \tag{12}$$

where *K* represents the conflict:

$$K = \sum_{B_1 \cap B_2 = \phi} m_1(B_1) \cdot m_2(B_2) \tag{13}$$

In this study, probabilistic classifiers are considered, so the masses (of the combination rule) are replaced by posterior probabilities [32].

#### 4.2.3. Final Decision Rule

Instead of providing a final decision on singleton hypotheses, and based on the highest probability rule as the decision criterion, the DS rule returns the final decision based on the overall expert responses.

The DS theory operation principle consists of both approving and consolidating; the proofs answer if they are in agreement or return an unfounded result that does not coincide with the desired target, in a significant conflict event between proofs. To overcome this problem, a post-processing process is proposed; the latter is applied to what is generated after merging outputs. Let *X* be the input data space and *Y* the output data space, where

each observation $x_i$ in $X$ is associated to labels of $Y$ with a posterior probability $\widetilde{h}_k$. The proposed $d_{\widetilde{h}}$ rule for post-processing outputs is as follows:

$$\forall\, x \in X \begin{cases} if\ \exists\, k\ \in\ [\![1,Q]\!]:\ \widetilde{h}_k(x) \geq 0.70,\ then\ d_{\widetilde{h}}(x) = k \\[2mm] else\ d_{\widetilde{h}}(x) = X^* \end{cases} \tag{14}$$

If one of the outputs is greater or equal to 0.70, the example is considered well classified; otherwise, the example is assigned to the reject class $X^*$. Indeed, it appears more logical and more prudent to reject an example with an uncertain assignment than to force an expert to give an answer through weighting biases.

The decision threshold has been set at 0.7, so that the gap between the dominant class and the next class always exceeds $0.4 = 1 - 0.7$. This threshold is judged quite sufficient and even severe, so that there is no confusion between classes. In other words, since the dominant class always wins a probability higher than 0.7, the eight other remaining classes only win together a probability lower than 0.3.

Furthermore, establishing the rejection class can return important information on the data quality and uncertainty/certainty (the data are erroneous and inconsistent). Indeed, a rejection is generally observed after the merger in two situations:

- Two or more classifiers are in conflict and do not agree on the class to which the example belongs, despite the fact that each classifier individually indicates a high probability of membership (the existence or not of a defect). In this case, the result provided by the classifiers lacks precision; and by classifying it as a rejection, the merger calls out the need to seek expert advice.
- The classifiers entirely agree on a dominant class absence and distribute the output probabilities in a uniform manner across several categories. In this case, the data reliability concerning the example is strongly called into question.

## 5. Results and Discussion

This section provides details on the used database, the M-SVM hyper parameter selection, the statistical evaluation parameters and the obtained results.

### 5.1. DGA Training, Validation and Evaluation Data

Dissolved Gas Analysis (DGA) is a technique developed to detect certain categories of incipient failures affecting oil-immersed equipment that cannot be easily detected by other conventional methods. This technique is considered one of the most used diagnostic and preventive monitoring tools today. The DGA database used in this study consists of 252 samples. Firstly, in the data collection part, 148 real mineral oil samples were collected from about 50 power transformers belonging to the Algerian utility "Sonelgaz Transport Electricity (STE)". The DGA database was interpreted by an expert in the field and subsequently subdivided into eight fault categories: Partial Discharge (PD: 16 samples), Low Energy Discharge (LED: 24 samples), High Energy Discharge (HED: 12 samples), Thermal fault ($t < 700\,°C$) (OH1: 16 samples), Thermal fault ($t > 700\,°C$) (OH2: 16 samples), Cellulose Degradation (CD: 20 samples), Thermal ($t > 700\,°C$) and Cellulose Degradation (OH2-CD: eight samples), Energy Discharge and Cellulose Degradation (ED-CD: 12 samples) plus a healthy samples category (N: 24 samples). Secondly, the lines that give the gas concentrations in an inequalities form (Table 4) are duplicated. A database containing 252 examples is therefore used for discriminator learning, validation and testing.

### 5.2. M-SVM Hyper Parameters Selection

In this study, four cross-validation levels were carried out. The database was firstly split randomly into four separate sets. Then, for each validation level and for each M-SVM model, two sets were used for training, one set was used for the best hyper parameter selection and one set was used for testing. Thus, all the database samples were considered during the training process and were tested during the evaluation process.

It is, of course, affirmative to validate an approach on all the samples than on a set portion (indeed, it could be that the results obtained with this portion are better than the results, which could be obtained with the other portions).

**Table 4.** Duplicating process for an ED-CD defect type.

| H$_2$ | CO | O$_2$ | N$_2$ | CO$_2$ | CH$_4$ | C$_2$H$_2$ | C$_2$H$_4$ | C$_2$H$_6$ |
|---|---|---|---|---|---|---|---|---|
| 127 | 847 | 16,973 | 89,517 | 3726 | 17 | 67 | <1 | 3 |
| | | | | $\Downarrow$ | | | | |
| 127 | 847 | 16,973 | 89,517 | 3726 | 17 | 67 | 0.2 | 3 |
| 127 | 847 | 16,973 | 89,517 | 3726 | 17 | 67 | 0.4 | 3 |
| 127 | 847 | 16,973 | 89,517 | 3726 | 17 | 67 | 0.6 | 3 |
| 127 | 847 | 16,973 | 89,517 | 3726 | 17 | 67 | 0.8 | 3 |
| 127 | 847 | 16,973 | 89,517 | 3726 | 17 | 67 | 1 | 3 |

The appropriate hyper parameter selection is a crucial step to build an efficient classification model. For an M-SVM type model, the hyper parameters to be optimized depend on the selected kernel type (in addition to the penalty parameters). The MSVMpack includes three kernels (linear, polynomial and radial) with which the experts can implement. In this study, the radial kernel, which allowed bringing out a more adequate separation, was used. For an M-SVM model selection (with radial kernel), two regularization parameters must be optimized among a grid of values: the kernel parameter $\gamma$ and the penalty parameter $C$. Thus, at each validation level, 255 combination pairs ($C$, $\gamma$) were evaluated with $\gamma$ and $C$ included respectively in the intervals $[2^4, 2^3, \ldots, 2^{-10}]$ and $[2^{12}, 2^{11}, \ldots, 2^{-2}]$ [37].

Then, each combination performance was reported by performing the learning on the training data and the evaluation on the validation data. The best combination was then retained for the final discrimination: the training was always done with the same learning data and the discrimination was based on the test data.

*5.3. Statistical Evaluation Parameters*

Seven statistical metrics are selected; all are deduced from a confusion matrix, Table 5, as follows:

- Receiver Operating Characteristic (ROC) curve: is a graphical representation that illustrates the classifier performances (Sensitivity and 1-Specificity) variation at all probability thresholds.
- Area Under the ROC Curve (AUROC-95\% CI): returns an overall performance estimate for all possible discrimination probability thresholds. A confidence interval is also associated to each measurement.
- Sensitivity and Specificity: these are two elementary and complementary metrics for the expert-performance evaluation. In fact, they are based on all the confusion matrix elements and constitute the ROC curve base.

**Table 5.** The confusion matrix.

| | | Predicted | |
|---|---|---|---|
| | | Considered category | Other categories |
| Reality | Considered category | True positive (TP) | False negative (FN) |
| | Other categories | False positive (FP) | True negative (TN) |

The sensitivity or True Positive Rate, $(\text{TPR}) = \dfrac{\text{TP}}{\text{TP} + \text{FN}}$, predicts the expert's ability to discern the positive population; while specificity or True Negative Rate, $(\text{TNR}) = \dfrac{\text{TN}}{\text{FP} + \text{TN}}$, predicts the expert's ability to discern the negative population.

- Positive Predictive Value (PPV): returns the truly positive individual proportion within a population classified as positive, $PPV = \dfrac{TP}{TP + FP}$.
- Negative Predictive Value (NPV): returns the truly negative individual proportion within a population classified as negative, $NPV = \dfrac{TN}{TN + FN}$.
- False Negatives (FN): is a positive population for which the test is negative, $FP = 1 - NR$.
- False Positives (FP): is a negative population for which the test is positive $FN = 1 - TPR$.

### 5.4. Obtained Results

The statistical performances of the proposed fusion method as well as those obtained by the implemented probabilistic direct M-SVM are shown in Tables 6 and 7 (for comparison). Also, Figures 5 and 6 reported the ROC curve evolution according to the five experts (for the NC class and the HED class). Focusing on these results, the following findings are reported:

**Table 6.** M-SVM and proposed DS-fusion model performances (all *p*-value < 0.001).

| Class | Statistical Parameters (%) | WW-MSVM | LLW-MSVM | MSVM$^2$ | CS-MSVM | Fusion |
|---|---|---|---|---|---|---|
| N | AUROC | 0.938 [0.883–0.993] | 0.963 [0.927–0.999] | 0.984 [0.969–0.999] | 0.991 [0.980–1.000] | 0.994 [0.985–1.000] |
| | Sensitivity | 87.5 | 90.62 | 93.75 | 100 | 100 (1 Reject) |
| | Specificity | 97.73 | 98.18 | 98.18 | 98.18 | 98.58 |
| | False positive | 2.27 | 1.82 | 1.82 | 1.82 | 1.42 |
| | False negative | 12.5 | 9.38 | 6.25 | 0 | 0 |
| | VPP | 84.85 | 87.88 | 88.24 | 88.89 | 91.18 |
| | VPN | 98.17 | 98.63 | 99.08 | 100 | 100 |
| PD | AUROC | 0.956 [0.901–1.000] | 0.970 [0.935–1.000] | 0.940 [0.896–1.000] | 0.975 [0.940–1.000] | 0.985 [0.960–1.000] |
| | Sensitivity | 90.62 | 87.50 | 87.50 | 90.62 | 93.54 (1 Reject) |
| | Specificity | 98.63 | 98.63 | 99.09 | 100 | 100 |
| | False positive | 1.37 | 1.37 | 0.91 | 0 | 0 |
| | False negative | 9.38 | 12.50 | 12.50 | 9.38 | 6.46 |
| | VPP | 90.63 | 90.32 | 93.33 | 100 | 100 |
| | VPN | 98.64 | 98.19 | 98.64 | 98.65 | 99.53 |
| LED | AUROC | 0.966 [0.937–0.996] | 0.959 [0.925–0.993] | 0.968 [0.941–0.996] | 0.968 [0.938–0.998] | 0.977 [0.949–1.000] |
| | Sensitivity | 87.5 | 93.75 | 90.62 | 90.62 | 90.32 (1 Reject) |
| | Specificity | 98.63 | 99.10 | 98.18 | 98.18 | 99.05 |
| | False positive | 1.37 | 0.90 | 1.82 | 2.82 | 0.95 |
| | False negative | 12.5 | 6.25 | 9.38 | 9.38 | 9.68 |
| | VPP | 90.32 | 93.75 | 87.88 | 87.88 | 93.33 |
| | VPN | 98.18 | 99.09 | 98.63 | 98.63 | 98.59 |
| HED | AUROC (95% CI) | 0.968 [0.925–1.000] | 0.971 [0.943–1.000] | 0.938 [0.867–1.000] | 0.958 [0.916–1.000] | 0.986 [0.970–1.000] |
| | Sensitivity | 83.33 | 87.5 | 83.33 | 79.16 | 86.96 (1 Reject) |
| | Specificity | 98.24 | 98.24 | 98.68 | 99.12 | 99.54 |
| | False positive | 1.76 | 1.76 | 1.32 | 0.88 | 0.46 |
| | False negative | 16.67 | 12.5 | 16.67 | 20.84 | 13.04 |
| | VPP | 83.33 | 84 | 86.96 | 90.47 | 95.24 |
| | VPN | 98.25 | 98.68 | 98.25 | 97.84 | 98.65 |
| OH1 | AUROC (95% CI) | 0.962 [0.931–0.993] | 0.971 [0.934–1.000] | 1.000 [1.000–1.000] | 0.996 [0.989–1.000] | 0.999 [0.996–1.000] |
| | Sensitivity | 89.28 | 92.85 | 100 | 96.42 | 96.42 |
| | Specificity | 98.66 | 100 | 100 | 100 | 99.53 |
| | False positive | 1.34 | 0 | 0 | 0 | 0.47 |
| | False negative | 10.72 | 7.15 | 0 | 3.58 | 3.58 |
| | VPP | 89.29 | 100 | 100 | 100 | 96.43 |
| | VPN | 98.66 | 99.12 | 100 | 99.56 | 99.53 |

**Table 6.** *Cont.*

| Class | Statistical Parameters (%) | WW-MSVM | LLW-MSVM | MSVM$^2$ | CS-MSVM | Fusion |
|---|---|---|---|---|---|---|
| OH2 | AUROC | 0.965 | 0.966 | 0.981 | 0.994 | 0.999 |
| | (95% CI) | [0.935–0.995] | [0.928–1.000] | [0.955–1.000] | [0.955–1.000] | [0.996–1.000] |
| | Sensitivity | 82.14 | 89.28 | 96.42 | 96.42 | 100 (2 Rejects) |
| | Specificity | 99.55 | 98.66 | 99.11 | 99.11 | 99.53 |
| | False positive | 0.45 | 1.34 | 0.89 | 0.89 | 0.47 |
| | False negative | 17.86 | 10.72 | 3.58 | 3.58 | 0 |
| | VPP | 95.83 | 89.29 | 93.10 | 93.10 | 96.30 |
| | VPN | 97.81 | 98.66 | 99.55 | 99.55 | 100 |
| CD | AUROC | 0.981 | 1.000 | 0.996 | 1.000 | 1.000 |
| | (95% CI) | [0.949–1.000] | [1.000–1.000] | [0.989–1.000] | [1.000–1.000] | [1.000–1.000] |
| | Sensitivity | 96.87 | 100 | 96.87 | 100 | 100 |
| | Specificity | 99.55 | 100 | 100 | 100 | 100 |
| | False positive | 0.45 | 0 | 0 | 0 | 0 |
| | False negative | 3.13 | 0 | 3.13 | 0 | 0 |
| | VPP | 96.88 | 100 | 100 | 100 | 100 |
| | VPN | 99.55 | 100 | 99.55 | 100 | 100 |
| OH2-CD | AUROC | 0.975 | 0.930 | 0.936 | 0.966 | 0.978 |
| | (95% CI) | [0.950–0.999] | [0.853–1.000] | [0.851–1.000] | [0.920–1.000] | [0.945–1.000] |
| | Sensitivity | 95 | 80 | 85 | 90 | 89.47 |
| | Specificity | 97.84 | 99.14 | 99.14 | 99.14 | 100 |
| | False positive | 2.16 | 0.86 | 0.86 | 0.86 | 0 |
| | False negative | 5 | 20 | 15 | 10 | 10.53 |
| | VPP | 79.17 | 88.89 | 89.47 | 90 | 100 |
| | VPN | 99.56 | 98.29 | 98.71 | 99.14 | 99.12 |
| ED-CD | AUROC | 0.965 | 0.966 | 0.932 | 0.959 | 0.972 |
| | (95% CI) | [0.920–1.000] | [0.936–0.996] | [0.847–1.000] | [0.916–1.000] | [0.926–1.000] |
| | Sensitivity | 83.33 | 91.66 | 87.50 | 87.50 | 100 (2 Rejects) |
| | Specificity | 98.68 | 97.80 | 98.68 | 98.25 | 98.64 |
| | False positive | 1.32 | 2.2 | 1.32 | 1.75 | 1.36 |
| | False negative | 16.67 | 8.34 | 12.50 | 12.50 | 0 |
| | VPP | 86.96 | 81.48 | 84 | 84 | 88 |
| | VPN | 98.25 | 99.11 | 98.68 | 98.68 | 100 |

**Table 7.** Average performances of M-SVM and DS-fusion models.

| | WW-MSVM | LLW-MSVM | MSVM$^2$ | CS-MSVM | Fusion |
|---|---|---|---|---|---|
| AUROC (95% CI) (*p*-value < 0.001) | 96.40 | 96.62 | 96.39 | 97.86 | 98.78 |
| Sensitivity | 88.39 | 90.35 | 91.22 | 92.30 | 95.19 |
| Specificity | 98.61 | 98.86 | 99.01 | 99.10 | 99.43 |
| False positive | 1.39 | 1.14 | 0.99 | 0.90 | 0.57 |
| False negative | 11.61 | 9.65 | 8.71 | 7.7 | 4.81 |
| VPP | 88.58 | 90.62 | 91.44 | 92.70 | 95.61 |
| VPN | 98.56 | 98.86 | 99.01 | 99.11 | 99.49 |

The expert performances increase proportionally with the amount of area under the curve; indeed, the expert whose predictions are correct tends to have an AUROC which is close to 100%; an expert whose predictions are erroneous has an AUROC which is close to 0%; and an expert whose prediction is not informative (random) has an AUCROC which is close to 50%. It can be seen from Table 7 that the obtained AUROC sets after applying DS fusion are superior to those of the four direct M-SVM implemented separately with an average AUROC of 98.78% (*p*-value < 0.001). This can be explained by the fact that the ROC curve plotting is based on the generalization rate variation at different probability thresholds. For the proposed fusion model, all examples are assigned to their categories with a probability greater than 0.70, while the four direct M-SVM employ an all-or-nothing classification; thus, in the worst case, an M-SVM can assign an example to a class with a probability of 0.12, which adversely affects the area under the curve.

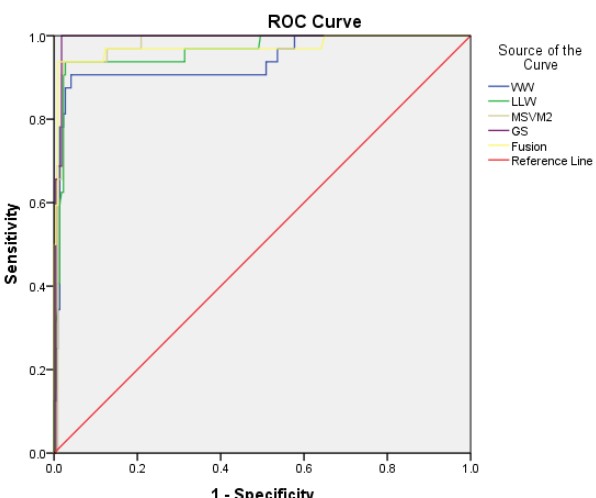

**Figure 5.** ROC curve evolution according to the five experts (class NC).

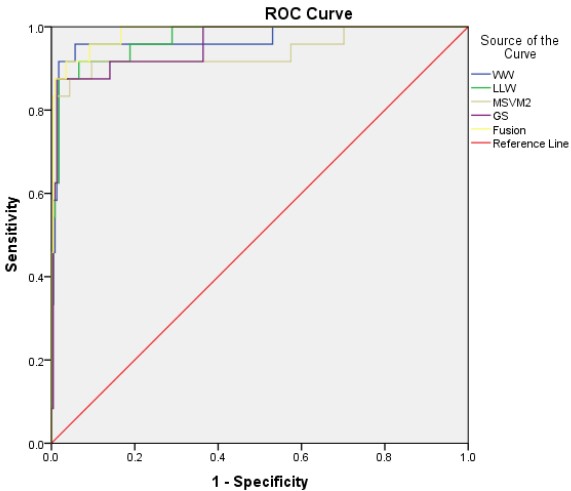

**Figure 6.** ROC curve evolution according to the five experts (class HED).

A confidence interval (significant at 95%) has also been associated with each AUROC to predict the area under the curve for the future independent samples (the generalization to a larger population); it was observed that the fusion AUROC upper limits can achieve an accuracy of 100% for all categories, contrary to M-SVM taken separately.

After merging, there is a significant distinction in the TPR and FN overall rates (which are considered to be correct classification rate estimators), just as the TNR and FP overall rates amelioration is underlined.

This is due, on the one hand, to the fact that the fusion takes into consideration each M-SVM advantage for a satisfactory result. Indeed, when analyzing the data reported in Table 6, it is found that it is difficult to privilege one model over another and that the M-SVM are in reality complementary (with slightly higher rates for CS-MSVM and slightly lower rates for WW-MSVM). For example, WW-MSVM tended to classify PD and OH2-CD categories well, while WW-MSVM generalizes the LED and HED classes well.

On the other hand, it should be remembered that a fusion outputs processing process was suggested, and a sample is considered to be well classified only if its membership probability is higher than 70%. Otherwise, it is reassigned to the reject category. Thus, attributing an example to the reject class instead of an arbitrary class favorably influences the results.

The FP and FN numbers (which are seen as indicators of overlapping rates between classes) increase consequently the model performances. In reliable diagnostic problems, it

is wiser to abstain and leave the choice to the experts in the field than to make a diagnosis whose consequences are irreversible.

After fusion, the results also return an important distinction of the PPV and NPV rates. This still adds a sure value to the proposed approach effectiveness, since these two statistical parameters are seen as being confidence indicators in the obtained results. For example, let us consider the proposed fusion model. The HED class PPV rate is 95.24%, which means that 95.24% of samples classified as HED are really HED, and that 4.76% of the samples are wrongly classified as HED; the HED class NPV rate is 98.65%, which means that 98.65% of samples classified as not HED are really not HED, and that 1.35% of the samples are wrongly classified as not HED.

According to the last column of Table 6, one can see that the rejected examples number after merging is eight examples out of 252, which represents a low percentage of the order of 3% (despite a very severe decision-making threshold). These results justify on the one hand that in 97% of cases the classifiers agree on the class to which the example belongs (which adds an important confidence degree to the results provided by our approach). On the other hand, the results obtained justify the dominant class presence and therefore validate the data reliability (please see Section 4.2.3).

At the end, a temporal analysis (study of the execution time) has been carried out. This proves the necessity to determine the possibility of introducing the proposed model into embedded processes. The analysis results are summarized in Table 8. Quite reasonable execution times for the four M-SVM and for the proposed fusion model can be observed.

**Table 8.** Average prediction times of M-SVM and DS-fusion models.

|  | WW-MSVM | LLW-MSVM | MSVM$^2$ | CS-MSVM | Fusion |
| --- | --- | --- | --- | --- | --- |
| Prediction time (s) | 0.018 ±0.004 | 0.015 ±0.002 | 0.010 ±0.003 | 0.014 ±0.004 | 0.085 ±0.009 |

However, a more important learning time was noticed. For learning diagnostic systems, the training phase performed offline and the time it may take is not important. What counts is the system response time in the operating phase. For the present study, the response time is almost instantaneous, which once again proves the efficiency of the proposed fusion approach.

*5.5. Comparison with Previous Work Results*

It is necessary to consider a comparison between the obtained results with the proposed approach and those obtained by other methods in the literature. In this subsection, a comparative study is proposed according to three parameters: the considered classes number, the used samples number and the overall generalization rate. For this, a few contributions whose treated problem is as similar as possible to that of the paper have been selected.

The authors proposed in their studies [7] a DGA diagnostic approach based on a Clustering Technique, a Cumulative Voting merged Technique (CVT) and a revisited k-nearest neighbors (KNN) algorithm. Xie and coauthors [12] proposed an approach based on relief algorithm for features selection, kernel Linear Discriminant Analysis (LDA) for features redundancy elimination and Relevance Vector machines (RVM) as discriminator. The authors in [14] proposed a system, namely: 2-ADOPT. This system was based on two versions of PSO approach (features selection and ensemble classifiers selection) and DS combination rule. The authors of [25] presented in their studies, an approach based on kernel extreme learning machine (KELM) optimized with the Harris hawks optimization (HHO) algorithm. The authors proposed in their research [27] a comparative study of an MLP network performance in power transformers diagnosis according to five different data sets. The study [30] proposed a new approach based on GA for features reduction and a krill herd (IKH) algorithm optimized SVM for generalization. Abdo et al. exposed in [31] a

method based on the application of a normalized data set (C-set) approach, an unsupervised Fuzzy C-means (FCM) clustering algorithm and a SVM. The study [32] proposes a multiple Probabilistic output Algorithms (PA) fusion (RVM, SVM and MLP) by the DS law; the hyper parameters of PA algorithms have been optimized by the PSO approach. From Table 9 it is noted that:

-	The categories number taken into account in this research is high compared to all other studies. Likewise, the present work offers a generalization rate which surpasses the obtained rates from other studies, except for the works [12,14,26]. This can be justified by the fact that the authors considered fewer classes and test examples.
-	The samples number is comparable to other studies with the exception of work [7]. This is justified by the fact that the latter considers a KNN type classifier, which necessarily needs a relatively large training sample for efficient performance. Contrary to this work, M-SVM-type discriminators are implemented, one of the advantages of which is to offer high generalization rates from a reduced learning sample.

**Table 9.** Previous work results.

| Study | Sample Count | Categories Number | Classification Rates (%) | | | | |
|---|---|---|---|---|---|---|---|
| CT-CVT-KNN [7] | 396 | 7 | 93 | | | | |
| SCA-RVM [12] | 135 | 6 | 97.07 | | | | |
| 2-ADOPT [14] | 101 | 4 | 97.94 | | | | |
| KELM-HHO [25] | 118 | 5 | 88 | | | | |
| MLP [27] | 102 | 6 | MLP-MRR | MLP-RR | MLP-IECR | MLP-KG | MLP-DR |
| | | | 87.88 | 90.91 | 93.94 | 100 | 90.91 |
| AG-IKH-SVM [30] | 113 | 5 | 85.71 | | | | |
| Cset-FCM-SVM [31] | 177 | 6 | 86.11 | | | | |
| PA-DS [32] | 156 | 6 | RVM | SVM | MLP | DS-Fusion | |
| | | | 87.8 | 85.3 | 82.6 | 89.1 | |
| Proposed Approach | 252 | 9 | WW | LLW | MSVM$^2$ | CS | Fusion |
| | | | 88.39 | 90.35 | 91.22 | 92.30 | 95.19 |

Especially since all the statistical tests that take into consideration the sample size crown with expressive areas under the curve (AUROC) and are statistically significant ($p$-values $< 0.001$), this testifies that the samples number is sufficient to validate the obtained generalization results.

Moreover, in these previous works, the authors took into account during the test only a limited part of the overall samples, while in the present study the samples were fully tested through the cross-validation application.

## 6. Conclusions

Accurate prediction and diagnosis of possible incipient faults is an indispensable part of power delivery to end-users, is conducive to sustainable development and has practical significance and demands. In this contribution, a power transformer intelligent diagnostic aid system based on real DGA data obtained from power transformers of a western Algerian electricity and gas company is proposed.

Six descriptor sets were firstly constructed and presented as Kohonen map inputs, for data size reduction and information completeness preservation. DS fusion was afterwards applied to the four Generic M-SVM model outputs for accurate and reliable decision-making. Finally, an output post-processing process was applied to overcome the contradictory evidence problem. It is worthwhile mentioning that the four direct M-SVM models implemented in this study offer theoretically the same performances, and each model has their own advantages and their own disadvantages; that the performance of MSV M was improved thanks to the proposed descriptor reconstruction approach; and that the deci-

sion making was strengthened through the application of DS fusion and the introduction of the rejection class.

Eight defect categories were considered in addition to the healthy samples class, and the obtained results highlighted the proposed approach effectiveness in detecting incipient failures. Indeed, the latter archieved an AUROC and sensitivity percentage of 98.78% and 95.19% (*p*-value < 0.001), respectively. Comparisons between the obtained results with different approaches reported in the literature [7,12,14,25,26,30,31] indicate a higher category number. Only a limited part of the overall samples was taken into account, while the samples were fully tested in the present contribution through the cross-validation application. In addition, the present work offers a generalization rate which surpasses the obtained rates from most studies.

Consequently, this study can be seen as very promising and useful for transformer owners. The results presented can be viewed as a benchmark and a challenge for further research. In particular, the authors' future research will focus on transformer anomaly prevention and exploration of other DGA data manipulation techniques such as ontologies.

**Author Contributions:** Conceptualization, M.H.; methodology, M.H.; software, M.H.; validation, M.H. and M.E.A.S.; formal analysis, M.H., M.E.A.S., F.M., M.B. and I.F; investigation, M.H.; resources, M.H.; data curation, M.H.; writing—original draft preparation, M.H.; writing—review and editing, M.H., I.F., F.M., M.B. and M.E.A.S.; supervision, M.B., I.F. and F.M.; project administration, M.B.; funding acquisition, M.H. and M.B. All authors have read and agreed to the published version of the manuscript.

**Funding:** This research has been supported by the PRFU-C00L07EP31022010001 project and the General Directorate for Scientific Research and Technological Development (DGRSDT).

**Institutional Review Board Statement:** Not applicable.

**Informed Consent Statement:** Not applicable.

**Data Availability Statement:** The database presented in this study includes real oil samples from around fifty power transformers belonging to the western Algerian electricity and gas company Sonelgaz Transport Electricity (STE). The considered equipment is of various ages and present a transformation ratio whose order is included in [60/10, 60/30, 220/60, 400/220] kV, a frequency of 50 Hz and a nominal power that varies between 10 and 120 MVA.

**Conflicts of Interest:** The authors declare no conflict of interest.

## Abbreviations

The following abbreviations are used in this manuscript:

| | |
|---|---|
| DGA | Dissolved Gas Analysis |
| M-SVM | Direct Multiclass Support Vector Machines |
| IA | Artificial Intelligence |
| D-T | Duval Triangle |
| R-R | Rogers Ratios |
| D-R | Dornenburg Ratios |
| IEC-R | IEC Ratios |
| KG | Key Gases |
| PTD | Power Transformers Diagnosis |
| G-P | Gases Percentage |
| TG | Total combustible and non-combustible Gases sum |
| TS | Total Sum |
| KSOM | Kohonen Self-Organizing Maps |
| GM-SVM | Generic M-SVM Model |
| WW | Weston and Watkins model |
| CS | Crammer and Singer model |
| LLW | Lee et al. model |
| M-SVM$^2$ | Quadratic Loss Multi-Class Support Vector Machine |
| DS | Dempster–Shafer fusion |

| | |
|---|---|
| PSO | Practical Swarm Optimization |
| GA | Genetic Algorithm |
| $H_2$ | Hydrogen |
| $O_2$ | Oxygen |
| $N_2$ | Nitrogen |
| CO | Carbon Monoxide |
| $CH_4$ | Methane |
| $CO_2$ | Carbon Dioxide |
| $C_2H_6$ | Ethane |
| $C_2H_4$ | Ethylene |
| $C_2H_2$ | Ethylene |
| VQ | Vector Quantification process |
| MSVMpack | MSVM software package |
| Bel | Belief function |
| Pl | Plausibility function |
| M | Mass function |
| ROC | Receiver Operating Characteristic curve |
| AUROC | Area Under the ROC Curve |
| TPR | True Positive Rate |
| TNR | True Negative Rate |
| PPV | Positive Predictive Value |
| NPV | Negative Predictive Value |
| FN | False Negatives |
| FP | False Positives |
| TP | True Positives |
| TN | True Negatives |
| STE | Sonelgaz Transport Electricity |
| PD | Partial Discharge |
| LED | Low Energy Discharge |
| HED | High Energy Discharge |
| OH1 | Thermal fault (t < 700 °C) |
| OH2 | Thermal fault (t > 700 °C) |
| CD | Cellulose Degradation |
| OH2-CD | Thermal (t > 700 °C) and Cellulose Degradation |
| ED-CD | Energy Discharge and Cellulose Degradation |
| N | Healthy samples |
| C-Set | Normalized data set approach |
| FCM | Fuzzy C-means clustering algorithm |
| KELM | Kernel extreme learning machine |
| HHO | Harris-Hawks-optimization algorithm |
| CVT | Cumulative Voting Technique merged |
| KNN | k-Nearest Neighbors algorithm |
| LDA | Linear Discriminant Analysis |
| RVM | Relevance Vector machines |
| MLP | Multilayer Perceptron |
| PA | Probabilistic output Algorithms |
| IKH | Krill herd algorithm |

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
