# Peer review of "Using Generic Direct M-SVM Model Improved by Kohonen Map and Dempster–Shafer Theory to Enhance Power Transformers Diagnostic"

_sustainability, doi:10.3390/su152115453_

Round 1

Reviewer 1 Report

This paper proposes a new intelligent system to improve the transformers diagnosis from DGA. However, some descriptions are not clear. Some revisions are necessary in the manuscript.

1. Please make sure that all parameters and abbreviations are defined.

2. Please add more about the mathematical model of the proposed algorithm.

3. The article uses many methods. Please further explain the relationship between them.

4. Please further explain the core innovation and significance of the article.

5. In the paper, authors have mentioned reliable operation of power transformers guarantees a good efficiency to the entire distribution network in complete safety. The significance of power installation to the safety of distribution network needs to be analyzed to indicate advantages of your work, which can refer to:

[a] IEEE Transactions on Industrial Informatics, vol. 18, no. 2, pp. 835-846, 2022

[b] IEEE Transactions on Industrial Informatics, vol. 19, no. 11, pp. 10751-10762, 2023

[c] Journal of Modern Power Systems and Clean Energy, vol. 11, no. 3, pp. 907-916, May 2023

[d] CSEE Journal of Power and Energy Systems, vol. 6, no. 3, pp. 681-692, Sept. 2020

Author Response

Dear Editor and reviewer,
We the authors are very grateful to the associate editor for his interest in our paper, and we also
thank, all the reviewers for their deep and in-depth comments. We have revised the manuscript
over the original version in light of the helpful reviewers suggestions and comments. We hope
that the revision has improved the document to a satisfactory level. Please find hereafter the
point-by-point responses to each reviewer’s queries and comments.
To Reviewer # 1 :
Note : modifications relating to your comments are mentioned in blue, and common
modifications with other correctors are noted in red.
This paper proposes a new intelligent system to improve the transformers diagnosis from DGA.
However, some descriptions are not clear. Some revisions are necessary in the manuscript.
Comment 1 : Please make sure that all parameters and abbreviations are defined.
Thank you for your comment; all observations and parameters have been checked one by
one. An abbreviations list can be found at the end of the paper. (lines: 733-735)
Comment 2 : Please add more about the mathematical model of the proposed algorithm.
Thanks for your suggestion. More details have been added in part 4.1.1, which relates to MSVM
Generic Model (GM-MSVM). (lines: 382-395)
Comment 3 : The article uses many methods. Please further explain the relationship between
them.
Comment 4 : Please further explain the core innovation and significance of the article.
Thank you for your suggestion. As a response to these two comments; in the introduction
section, just before the article organization, the following paragraph is added: (lines: 121-
130)
Thus, the proposed system has a primary objective to improve descriptor extraction step
with the proposed KSOM parameters reconstruction approach. The latter effectively
minimizes the information loss as much as possible (unlike the traditional functionalities
selection approaches). The second objective focus on implementing the four direct MSVM
through the Generic M-SVM Model, with the aim of overcoming the limits of
traditional MSVM based on decomposition methods (minimize the classifier complexity
and save execution time). Finally, the proposed system has the final objective of
strengthening final decision-making by merging four M-SVM by DS Fusion; after, an
approach to solve the contradictory evidence problem, linked to DS fusion is proposed.
The section 2 describes in detail the motivations and originality of the approach taken.
Comment 5 : In the paper, authors have mentioned reliable operation of power transformers
guarantees a good efficiency to the entire distribution network in complete safety. The
significance of power installation to the safety of distribution network needs to be analyzed to
indicate advantages of your work, which can refer to:
[a] IEEE Transactions on Industrial Informatics, vol. 18, no. 2, pp. 835-846, 2022
[b] IEEE Transactions on Industrial Informatics, vol. 19, no. 11, pp. 10751-10762, 2023
[c] Journal of Modern Power Systems and Clean Energy, vol. 11, no. 3, pp. 907-916, May
2023
[d] CSEE Journal of Power and Energy Systems, vol. 6, no. 3, pp. 681-692, Sept. 2020
Thanks for your suggestion, the content of the introduction first paragraph has been
modified (the paragraph noted in red has been added and two suggested bibliographies
have been added to the references list). (lines: 36-43)
“Indicators of sustainability focusing on energy are crucial tools used to assess and
monitor progress toward guaranteeing electricity delivery to end-users [1]. In the last
decades, power grids are facing growing interests for deploying new and intelligent
technology for improved reliability and availability of power supply. This is important in
meeting new challenges due to accelerating urbanization and evolving requirements to
ensure smart cities. The smart city concept mostly relies on cameras, sensors and
monitoring tools to maintain or support human well-being continuously over time. The
data collected is processed and analyzed to improve operational efficiency of major
equipment such as power transformers, public safety, and life quality, and also ensure
efficient electrical installation [a, b]. In this context, accurate monitoring of major assets
is essential. Power transformers, which are the most essential and expensive devices in
the power transmission and distribution networks, are aging worldwide ahead of their
theoretical design life. Their role is to facilitate the transition between the different
electrical network levels (production, transport and distribution), by recommending an
adapted voltage to them (defer the electrical input quantities into quantities of different
values) and ensuring the connection between them (while minimizing losses by joule
effects). Due to their importance in an electrical structure, their reliable operation
guarantees a good efficiency to the entire distribution network, and ensures them an
economic gain. Indeed, when a power transformer breaks down due to various factors, it
effects, the generator output which will cause significant damage to the electric
companies economy and to the users property. Thus, it is necessary to ensure an excellent
monitoring of the power transformers health state in order to avoid sudden breakdowns
[2-3].”

Reviewer 2 Report

This article reports a very interesting use with some benchmarking of combined Kohonen Self-Organizing Maps with Genetic / M-SVM algorithm to improve operational efficiency assessment using DGA for important equipment.

I have identified a few items to target further the studies’ outcomes and recommendations to translate them into material clarification and potential recommendations for future research.

Section 1 Introduction, 3rd paragraph, p. 2 [Lines 66-67]: Please be more accurate (or add an explanation) on why “…the different approaches sometimes give contradictory answers for the same sample…”. Is it because of errors in coding, sensitivity, or (likely) other reasons? Explaining why using machine learning with SVM or other tools could solve such contradictions would be helpful in understanding the roots of the research.

Section 1 Introduction, 4th paragraph, p. 2 [Lines 85] and Figure 1 p. 4: Please clarify why normalization (rather than Standardization) was used for Feature scaling. It is one of the most critical steps during data pre-processing before creating a machine-learning model with such “sensitive” datasets. Normalization is used when we want to bound our values between two numbers (e.g., between [0,1] or [-1,1]), while Standardization transforms the data to have zero mean and a variance of 1. The latter would make data unitless. Scaling can make a difference between a weak machine-learning model and a better one. This would need clarification in the Introduction section (presenting the block diagram of the PRD), or in Section 3.3. Features normalization

Section 2. Study motivations and innovations, p. 4 [Lines 139-142]: Please outline the main difference(s) between the works of Yiyi et al. [27], both using KSOM and Genetic Algorithm and the authors’ work. Is it on improving the execution time? Is it specifically relying on the classification and M-SVM integration, or are there other important items? They would deserve more explanation on how they differentiate or validate using the combined approach (as it seems to be referred to in Lines 171-172). If pertinent, it could also be clarified in Section 2 or Section 3.4 (explaining the specifics of this work compared with [27] in using the Kohonen method). These items could prove helpful for the readers.

Section 6. Conclusions. Lines 617-619 refer to a definition. In conclusion, we shouldn’t find a conventional definition.  The DGA definition is not a new definition and should be put in section 5.1 (discussion DGA data) or else: “Dissolved Gas Analysis (DGA) is a technique developed to detect certain categories of incipient failures affecting oil-immersed equipment that cannot be easily, detected by other conventional methods.”  On the other hand, there is a small conclusion mentioned in section 2 that could be recalled in the Conclusions section (i.e., Line 186-188: “…the four direct M-SVM models implemented in this study offer theoretically the same performances, and each model has their own advantages and their own disadvantages”), since it refers to some of the studies outcomes.

Section 5.3. Statistical evaluation parameters, Lines 487-492. For clarity, these 2 sentences on “Sensitivity and Specificity” should be kept in the same bullet point since they are talking about the same metrics TPR and TNR.

In the Conclusions section, Line 611. Please clarify why coming with this example towards the end of the article when it’s not referred to before. “The smart city concept mostly relies on cameras, sensors and monitoring tools to maintain or support human well-being continuously over time.” Also, the Conclusions would benefit from adding a brief version of some partial conclusions in Section 5.5, notably touching on the interesting comparison between the obtained results with the proposed approach and those obtained with other methods in the literature (notably around Lines 594-602).

The language quality is generally clear and sound. Some typos that a quick spell-check could resolve are present. Notably, several “spaces” in excess or missing in the text could be easily corrected, e.g., Lines 22, 205, 226, 255, 304, 340, 416, 480, 510, 557, etc.

Lines 208. Needs an adjustment of the format for clarity, either adding this as an equation format or making sure it is well integrated into the text Σ(H2, O2, N2, …)

Section 4.3.2. Line 365. It seems the sentence is cut halfway (“Thus, it is desirable and more reliable to consider classifiers that”) and makes no sense.  But it might be integrated into the following paragraph, starting [Line 368] with another incomplete sentence: “Return a posteriori probability estimates, instead of a deterministic classification (which returns the membership class label).” Anyway, this should be reviewed, adding why it is “desirable and more reliable” from the authors' standpoint.

Lines 489 and 490.  These two sentences should be kept in the same bullet point, “Sensitivity and Specificity,” for more clarity.

Author Response

Dear Editor and reviewer,

We the authors are very grateful to the associate editor for his interest in our paper, and we also thank, all the reviewers for their deep and in-depth comments. We have revised the manuscript over the original version in light of the helpful reviewers suggestions and comments. We hope that the revision has
improved the document to a satisfactory level. Please find hereafter the point-by-point responses to each reviewer’s queries and comments.

with our very best regards

Reviewer 3 Report

The subject of the study is related to issues related to energy networks, the key elements of which are power transformers ensuring reliability and sustainable energy. After reading the content of the study, I suggest the authors organize and supplement the content in subsections following the comments below:

1. The abstract should be supplemented with the purpose of the article.

2. The Introduction chapter should be divided into a part about transformers and the proposed research approach. Moreover, it should be justified why the research approach proposed by the authors is better than competitive ones.

3. In the introduction, it should be indicated why the topic undertaken by the authors is important and how the research/analyses conducted have expanded knowledge about the problem.

4. Chapter 2 should be supplemented with research hypotheses that should be verified in the research results.

5. Check the correctness of the calculations.

6. The article should be supplemented with a discussion chapter.

7. The Conclusion chapter should be supplemented with the limitations of the research conducted, as well as the implications for people managing energy supplies.

8. In the Conclusions chapter, you should describe the authors' further research plans and indicate to whom the research results are addressed.

9. The English language requires correction. The text is mainly missing punctuation marks. Additionally, you should eliminate typos and use the right number of words.

The English language requires correction. The text is mainly missing punctuation marks. Additionally, you should eliminate typos and use the right number of words.

Author Response

(The authors gave the same response as above.)

Round 2

Reviewer 1 Report

The paper has been revised and is recommended for acceptance